# Modeling Wildland Firefighters' Assessments of Structure Defensibility

**Alexander J. Heeren** [1],*, **Philip E. Dennison** [1], **Michael J. Campbell** [1] **and Matthew P. Thompson** [2]

1   Department of Geography, University of Utah, Salt Lake City, UT 84112, USA;
    dennison@geog.utah.edu (P.E.D.); mickey.campbell@geog.utah.edu (M.J.C.)
2   US Department of Agriculture Forest Service, Rocky Mountain Research Station, Fort Collins, CO 80526, USA;
    matthew.p.thompson@usda.gov
*   Correspondence: alex.heeren@utah.edu

**Abstract:** In wildland–urban interface areas, firefighters balance wildfire suppression and structure protection. These tasks are often performed under resource limitations, especially when many structures are at risk. To address this problem, wildland firefighters employ a process called "structure triage" to prioritize structure protection based on perceived defensibility. Using a dataset containing triage assessments of thousands of structures within the Western US, we developed a machine learning model that can improve the understanding of factors contributing to assessed structure defensibility. Our random forest models utilized variables collected by wildland firefighters, including structural characteristics and the surrounding ignition zone. The models also used landscape variables not contained within the triage dataset that captured important information about accessibility, vegetation, topography, and structure density. We achieved a high overall accuracy (77.8%) in classifying structures as defensible or non-defensible. The presence of a safety zone was the most important factor in determining structure defensibility. Road proximity, vegetation composition, and topography were also found to have high importance. In addition to improving the understanding of factors considered by wildland firefighters, communities could also gain from this information by enhancing their wildfire response plans, focusing on targeted mitigation, and improving their overall preparedness.

**Keywords:** structure triage; structure defensibility; safety zone; wildland urban interface (WUI); machine learning; firefighter decision-making



## 1. Introduction

Wildland fires can threaten communities, potentially causing widespread property destruction and loss of life [1]. Historical land management practices and anthropogenic climate change have increased fire activity in the Western US [2–4], where the development of homes and infrastructure in the wildland–urban interface (WUI) has further elevated wildfire risk to people [5,6]. The WUI is highly prone to wildland fires and presents considerable risks to human lives and property [7,8]. Approximately 69% of structures destroyed by wildfires in the conterminous US are located in WUI areas [6]. Moreover, the WUI is among the fastest-growing land-use types in the US, witnessing a 41% increase in new homes built between 1990 and 2010 [7].

Due to persistently high fire danger and continued development in WUI areas, protecting structures from wildland fires in the WUI remains extremely challenging [9,10]. Firefighters not only have to try to suppress the fire but also protect structures. The situation becomes even more daunting when faced with resource limitations, particularly on incidents where a significant number of structures are at risk [11]. Structure protection also presents challenges to wildland firefighter safety. Wildfires in the WUI give rise to exceptionally hazardous and volatile conditions, demanding quick, critical decision-making from firefighters and other emergency responders.

To address these challenges, wildland firefighters in the US employ a systematic approach known as "structure triage", where trained personnel assess structures during major wildland fire incidents. This process uses a standardized method of rapidly identifying, prioritizing, and categorizing structures according to their defensibility, following specific guidelines encompassing firefighter safety, fire behavior, structure characteristics, tactical challenges and hazards, and structure protection strategies [12]. In the context of wildland fire management, defensibility refers to whether or not a structure can safely and successfully be defended against damage or loss due to fire.

For structure triage, firefighters conduct on-the-ground visual assessments of each structure. They determine whether sufficient escape routes and safety zones exist where a threatened firefighter can find adequate refuge from dangerous fire conditions [13–15]. Next, they visually inspect the Home Ignition Zone (HIZ), a combination of the structure's building materials/design (i.e., roofing and siding, vents, decks, windows, and eaves), as well as the structure's immediate surroundings within 100 feet (i.e., available fuels, vegetation management, and topography) [3], and determine any necessary defense measures. Structures are more likely to be deemed defensible if they have a low risk of ignition, such as those built with fire-resistant materials or having adequate spacing from nearby vegetation. Typically, these structures are prioritized for defense over structures with a higher probability of ignition because they can be protected using fewer resources and less time. The assessments are entered into an electronic form provided by the National Interagency Fire Center (NIFC). This information serves as a valuable resource for fire incident management and operation personnel, enabling decision-makers to effectively allocate resources, prioritize structure protection efforts, and coordinate actions with personnel involved in fire operations.

Past research has shown that factors like building materials, design, topography, the surrounding vegetation and defensible space, greatly impact a structure's survival [9,16–24]. Recent studies examining features at both the landscape scale (covering a large spatial extent) and local scale (focused on areas close to the point of interest) indicated that landscape scale vegetation is a more reliable indicator of structure loss than local scale [25,26]. Structure density may also contribute to survivability. Past work has demonstrated a higher risk of loss in areas with higher structure densities, suggesting that close proximity and structure-to-structure ignition play a significant role [1,17,27–29]. Conversely, other studies propose a greater risk of loss in lower-density areas due to their proximity to wildland vegetation [22,23,25,30]. Accessibility may affect a structure's survival and wildland firefighter safety. Narrow or dead-end roads increase risks for firefighters by limiting escape routes and impeding firefighting efforts [31,32]. In contrast, a wider, well-connected road system could improve fire engine access and decrease the risk of structure loss. Syphard et al. [30] found that isolated housing clusters with limited road access were more prone to structure loss.

Despite extensive research on factors contributing to structure loss, the strategies that firefighters employ to assess structure defensibility and the factors that may influence their decision-making remain unexplored. This research aims to develop predictive models that improve the understanding of firefighter structure triage during wildfires. By leveraging a large, rich structure triage dataset, augmented with a host of GIS-derived landscape characteristics, and a random forests machine learning algorithm, we seek to better understand the driving forces behind firefighters' interpretations of defensibility.

## 2. Materials and Methods

### 2.1. Structure Triage Dataset

The structure triage dataset used in this study was provided by NIFC [33]. This dataset became operational during the 2020 fire season, and remains a working dataset to which new data are being collected, updated, and accumulated on active incidents where structures are threatened by wildland fire. We used structure assessments collected

over a period spanning 20 June 2020 through 8 October 2021 in our analysis [33]. To our knowledge, this is the first published analysis of the structure triage dataset.

Our study area consisted of 11 western states of the contiguous US: Arizona, California, Colorado, Idaho, Montana, Nevada, New Mexico, Oregon, Utah, Washington, and Wyoming (Figure 1). Although the structure triage dataset has national coverage, 96% of all the records fall within these western states. Using the few records in the eastern US and Alaska could bias or weaken a model aimed at predicting structure defensibility due to large geographical distances, different wildland fire practices, and fire regimes. Therefore, the structure triage data points outside the 11 Western US states were excluded from the analysis.

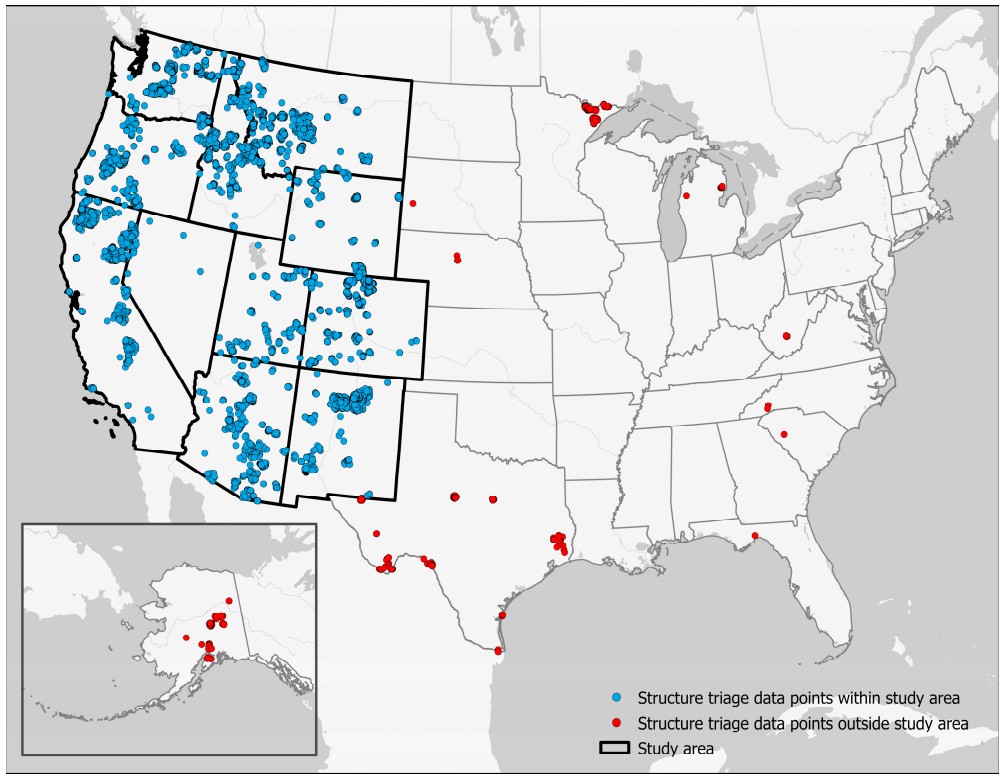

**Figure 1.** Study area map displaying the structure triage data points utilized for analysis, as well as those that were excluded. Base map credit: Esri.

Structure triage category is the variable that encompasses the four categories of structure defensibility determined by wildland firefighters, and was used as the target or response variable in this analysis. These four categories were as follows: (1) Defensible–Stand-Alone (D-SA); (2) Defensible–Prep and Hold (D-PH); (3) Non-Defensible–Prep and Leave (ND-PL); and (4) Non-Defensible Rescue Drive-By (ND-RDB) (Figure 2). D-SA structures need no work to improve defensibility and are likely to stand independently against the fire. Structures deemed D-PH are considered defensible, with little wildland firefighter intervention. They are safe enough for crews to stay and implement structure protection tactics as flames approach. ND-PL structures are not likely to survive a fire without any mitigation measures. Since these structures are unsafe for crews, they will perform mitigation measures as time allows and will pull back as the flames near, returning after the main fire front passes to implement structure protection tactics. ND-RDB are those considered too unsafe for firefighters to work on, and no amount of mitigation measures can be taken to ensure a structure's survival [12].

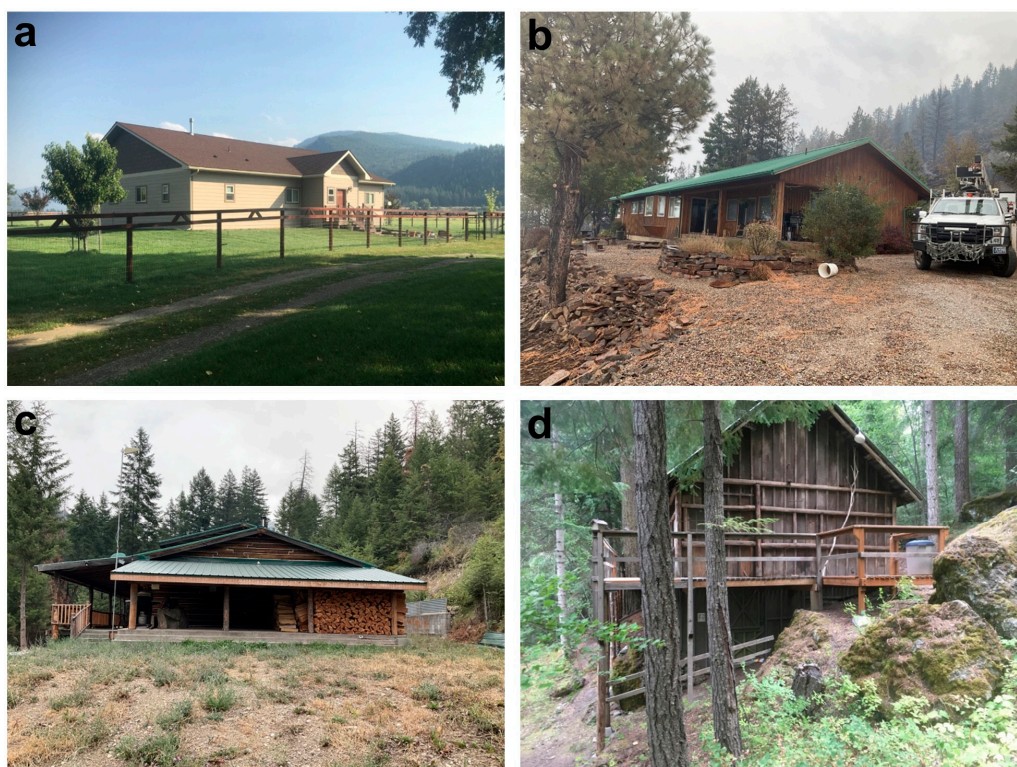

**Figure 2.** Structure triage categories defined by wildland firefighters during structure triage assessment. Images were obtained from the structure triage dataset [33]. (**a**) A D-SA structure, with vinyl siding and minimal vegetation in close proximity, which exhibits a very low probability of ignition. (**b**) A D-PH structure that has adequate defensible space and accessibility but is located on a slope, which may necessitate additional protection tactics. (**c**) An ND-PL structure which is a log cabin with firewood stacked against the structure, resulting in a substantial risk of ignition. However, it benefits from having minimal vegetation in its immediate vicinity. (**d**) An ND-RDB structure that presents several concerns, including the presence of overhanging trees, its positioning on a slope, and the use of wood siding, all of which increase the risk of ignition.

*2.2. Analysis Variables*

Analysis variables were selected based on variables included in the structure triage dataset, plus additional GIS-derived variables describing vegetation, topography, structure density, and road characteristics. Previous work has demonstrated relationships between similar GIS-derived variables and structure loss [16,17,25,26,30]. The structure triage dataset includes several variables related to structure defensibility that were used in modeling (Figure 3). *Structure Type* refers to whether the structure was residential, outbuilding, or commercial-type. *Access* describes what type of wildland fire resource could safely enter a property and efficiently protect a structure (i.e., type 1 engine, type 3 or 4 engine, type 6 engine, handcrew, or other). *Water Supply* may or may not be present or reliable in a wildland fire situation; this variable states what type of water access was near a structure, whether fire hydrant(s), sprinkler system, or no reliable water source nearby. A *Safety Zone* variable indicates whether firefighters determined there was an adequate safety zone within a given area or if a safety zone was absent. A safety zone is a location where a threatened firefighter can find adequate refuge from an approaching fire [13]. *Defensible Space* was categorized based on the distance of modified vegetation and fuels from a structure, including categories of no defensible space, 30 feet (9.1 m), greater than 30 feet, or greater than 100 feet (30.5 m) of defensible space. *Fuel Type* indicates the dominant type of vegetation, describing grass, shrub, or timber present adjacent to a structure. *Roof Materials* and *Siding Materials* were categorized by different material types (i.e., wood, rock, concrete, metal, wood, or fire-resistant materials) and their degree of combustibility. *Electricity to a*

*Structure* and *Gas to a Structure* were noted because they can pose a dangerous hazard to a firefighter. *Primary Structures* specify the number of dwellings where people predominantly live on the property. Outbuildings indicate the number of structures surrounding the primary structure(s) that are not inhabited. Incident command is the on-scene incident command and/or management organization where command, control, and coordination of an emergency response is located. *Communication with Incident Command* indicated whether a secure line of communication can be established and maintained from the structure location. Whether or not a building is occupied provides information to firefighters to gauge life safety risks. *Mitigation Hours* is an estimated time that defensive measures will be require to lower the probability of the structure igniting. Finally, the structure triage dataset includes *Latitude* and *Longitude* indicating the location of each structure.

**Structure Triage Variables** / **GIS-Derived Variables**

| Variable | | | | | |
|---|---|---|---|---|---|
| Structure Type | EVT LF 30 m | EVT ORDER 60 m | EVH 90 m | EVC Herb 6 km | Road Density 30 m |
| Access | EVT LF 60 m | EVT ORDER 90 m | EVH 360 m | Slope 30 m | Road Density 60 m |
| Water Supply | EVT LF 90 m | EVT ORDER 360 m | EVH 600 m | Slope 60 m | Road Density 90 m |
| Safety Zone | EVT LF 360 m | EVT ORDER 600 m | EVH 900 m | Slope 90 m | Road Density 360 m |
| Defensible Space | EVT LF 600 m | EVT ORDER 900 m | EVH 3.6 km | Slope 360 m | Road Density 600 m |
| Fuel Type | EVT LF 900 m | EVT ORDER 3.6 km | EVH 6 km | Slope 600 m | Road Density 900 m |
| Roof Material | EVT LF 3.6 km | EVT ORDER 6 km | EVC Shrub 600 m | Slope 900 m | Road Density 3.6 km |
| Siding Material | EVT LF 6 km | EVT CLASS 30 m | EVC Shrub 900 m | Slope 3.6 km | Road Density 6 km |
| Electricity to Structure | EVT PHYS 30 m | EVT CLASS 60 m | EVC Shrub 3.6 km | Slope 6 km | Closest Major Road Type |
| Gas to Structure | EVT PHYS 60 m | EVT CLASS 90 m | EVC Shrub 6 km | Structure Density 30 m | Distance to Major Road Type |
| Primary Structures | EVT PHYS 90 m | EVT CLASS 360 m | EVC Tree 600 m | Structure Density 60 m | Closest Road Type |
| Outbuildings | EVT PHYS 360 m | EVT CLASS 600 m | EVC Tree 900 m | Structure Density 90 m | Distance to Closest Road Type |
| Communication with Incident Command | EVT PHYS 600 m | EVT CLASS 900 m | EVC Tree 3.6 km | Structure Density 360 m | |
| Building Occupied | EVT PHYS 900 m | EVT CLASS 3.6 km | EVC Tree 6 km | Structure Density 600 m | |
| Mitigation Hours | EVT PHYS 3.6 km | EVT CLASS 6 km | EVC Herb 600 m | Structure Density 900 m | |
| Latitude | EVT PHYS 6 km | EVH 30 m | EVC Herb 900 m | Structure Density 3.6 km | |
| Longitude | EVT ORDER 30 m | EVH 60 m | EVC Herb 3.6 km | Structure Density 6 km | |

**Figure 3.** List of structure triage and GIS-derived predictor variables.

Although the structure triage data contained an array of useful information, we sought to determine if additional GIS-derived variables could augment the model dataset and provide additional predictive or explanatory capacity. To that end, we incorporated additional variables derived from publicly available geospatial datasets, including vegetation information, topography, structure density, and accessibility (Figure 3). This broadened the scope of our analysis and offered a more comprehensive view of other factors potentially influencing structure assessments not included in the triage data. Vegetation data offer insights into fuel characteristics, including fuel type and density, which directly impact the spread and intensity of wildfires. Topography, which also plays a critical role in dictating fire behavior, sheds light on terrain effects on defensibility. Structure density helps identify where higher or lower density may affect the assessment of defensibility. Accessibility variables provide insight into efficient road access to structures. For all of these additional

variables, each structure triage data point was evaluated using summary statistics derived from varying radii (30 m, 60 m, 90 m, 360 m, 600 m, 900 m, 3.6 km, and 6 km) surrounding the structure, allowing us to capture both local scale (i.e., the HIZ), and broader landscape scale perspectives.

For the vegetation variables, we used data from LANDFIRE. The US LANDFIRE program provides geospatial products, including vegetation, wildland fuels, and fire regimes, at a national scale with a 30 m resolution raster format [34]. We employed LANDFIRE's Existing Vegetation Layers, which encompass *existing vegetation type* (EVT), *existing vegetation height* (EVH), and *existing vegetation cover* (EVC) [35]. EVT is a product for which each raster pixel represents the dominant plant community type. For our analysis, variables that describe the vegetation physiognomy were included, such as vegetation lifeform (EVT LF), vegetation physiognomy (EVT PHYS), vegetation physiognomic order (EVT ORDER), and vegetation physiognomic class (EVT CLASS). Using a majority focal statistic operation, the most common vegetation types across all pixels within circular areas surrounding each structure were extracted at a range of radii (30 m, 60 m, 90 m, 360 m, 600 m, 900 m, 3.6 km, and 6 km) (Figure 3).

EVH is a product for which each pixel represents the average height of the dominant plant lifeform (i.e., trees, shrubs, or herbaceous). Due to height dependence on lifeform, the original categorical height classification scheme of EVH was replaced with a continuous representation, where the distinction between plant lifeforms was removed. With this change, each pixel solely represented the estimated height of vegetation in meters, providing a unified measure of vegetation height across the entire dataset. A mean statistical operation was used to extract the average vegetation height value of pixels within each radius (30 m, 60 m, 90 m, 360 m, 600 m, 900 m, 3.6 km, and 6 km) (Figure 3).

EVC originally represented the percent canopy cover of dominant plant lifeforms (trees, shrubs, or herbaceous) per pixel. To streamline our analysis and account for how each lifeform contributes as fuel for wildland fires, we transformed EVC's categorical cover classification into three continuous representations, each corresponding to a specific lifeform's cover as a percentage. In this representation, each pixel represents the dominant lifeform's cover percentage, while pixels not specific to that lifeform were assigned "NA" (not applicable) values. As all three lifeforms are less likely to be in proximity to structures, leading to missing values when extracting data especially at smaller radii, we concentrated on extracting the mean cover for all three plant lifeforms from radii of 600 m, 900 m, 3.6 km, and 6 km (Figure 3).

We also utilized the LANDFIRE 2020 Slope Degree product, which indicates the steepness of surface inclines in degrees, ranging from 0 to 90 per pixel [36]. The mean statistical operation was employed to extract the average slope value from pixels within radii of 30 m, 60 m, 90 m, 360 m, 600 m, 900 m, 3.6 km, and 6 km (Figure 3).

Structure density was calculated from Microsoft's deep learning-generated building footprints dataset, which provides polygons for each structure across the entire US [37]. We transformed these polygons into points and then created multiple density rasters, each capturing the density of structures per square kilometer within 30 m, 60 m, 90 m, 360 m, 600 m, 900 m, 3.6 km, and 6 km radii (Figure 3). These rasters enabled us to extract density values at each structure triage data point.

We also incorporated accessibility into our analysis using the United States Census Bureau's Topologically Integrated Geographic Encoding and Referencing system (TIGER) dataset. This dataset contains detailed road type attributes and line features representing streets and roads [38]. We derived multiple variables from this dataset, including road density at varying radii, distance to major roads, the closest major road type to each structure triage data point, and any road type nearest to a structure triage data point (Figure 3).

*2.3. Spatial Partitioning*

The structure triage dataset has an underlying issue of spatial autocorrelation [39] that can be attributed to large numbers of structures in close proximity being assessed when threatened by wildfire. Structures within a neighborhood or community may be more likely to have similar structure triage attributes. Spatial autocorrelation issues arise when validating machine learning models such as random forests [40]. In a random sample, neighboring structures or many structures in the same neighborhood or community could end up in both the test and training datasets (Figure 4a). This would violate the model assumption that observations are independent of one another by having observations close together with similar values in both sets, resulting in inflated model accuracy.

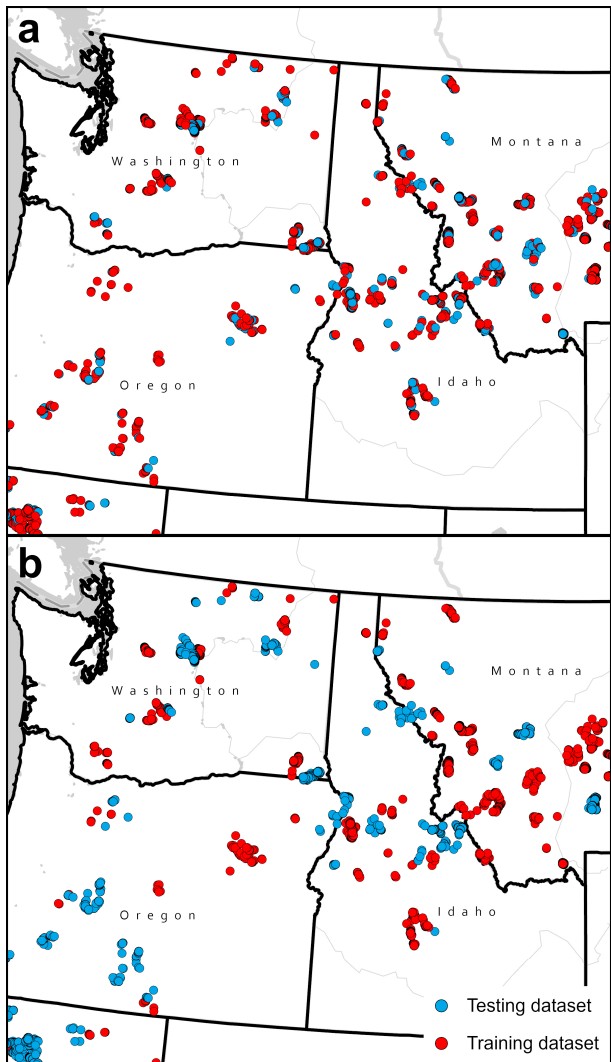

**Figure 4.** Testing and training structure triage data points for a subset of the study area. (**a**) Typical random forest modeling randomly splits observations into training and test datasets through random sampling. This leads to structures from the same neighborhood or community being included in both the testing and training datasets. (**b**) We used spatial partitioning where each observation is grouped by the fire it was threatened by. Base map credit: Esri.

To reduce potential impacts of spatial autocorrelation, a method known as "spatial partitioning" was employed [41]. Each structure triage data point was grouped according to the specific wildfire it faced, using the fire's assigned name as an identifier (Figure 4b). Observations were linked to the corresponding fire based on the dates of data collection and active fire periods, along with visual comparisons of locations and proximity

of past fire perimeters from the Wildland Fire Interagency Geospatial Services (WFIGS) Group's dataset, which consolidates certified interagency fire perimeters from 2020 to the present [42]. If a fire name could not be associated with a structure triage data point, it was excluded from the analysis.

The partition function [43] from the R package Groupdata2 version 2.0.2 was used to divide the structure triage data points into testing (30%) and training (70%) datasets. This function ensures a balanced distribution by grouping data points based on a specified categorical variable, in this case fire name.

### 2.4. Modeling

Random forests is a machine learning method that constructs a collection of decision trees by building each tree with a random subset of the training data, and the predictions from these trees are aggregated to produce a final prediction [44]. This method was ideal for this analysis for several reasons, including supporting both binary and multi-class classification tasks, the capability to handle both categorical and continuous predictors, being able to characterize non-linear and interactive relationships between predictor and response variables, and providing robust measures of variable importance. We utilized the random forest classification algorithm as implemented in the ranger package version 0.14.1 [45] in R version 4.2.0, [46]. To optimize the performance of each model, we leveraged the tuneRanger package version 0.5 [47] for model tuning.

This framework enabled us to create and evaluate two classification models for wildland firefighter assessments of structure defensibility: (1) a two-class classification model in which we reduced the four categories of structure triage (D-SA, D-PH, ND-PL, and ND-RDB) to two categories, where D-SA and D-PH were collapsed into a *defensible* category and ND-PL and ND-RDB were collapsed into a *non-defensible* category; and (2) a four-class classification model with all four structure triage categories (Figure 5).

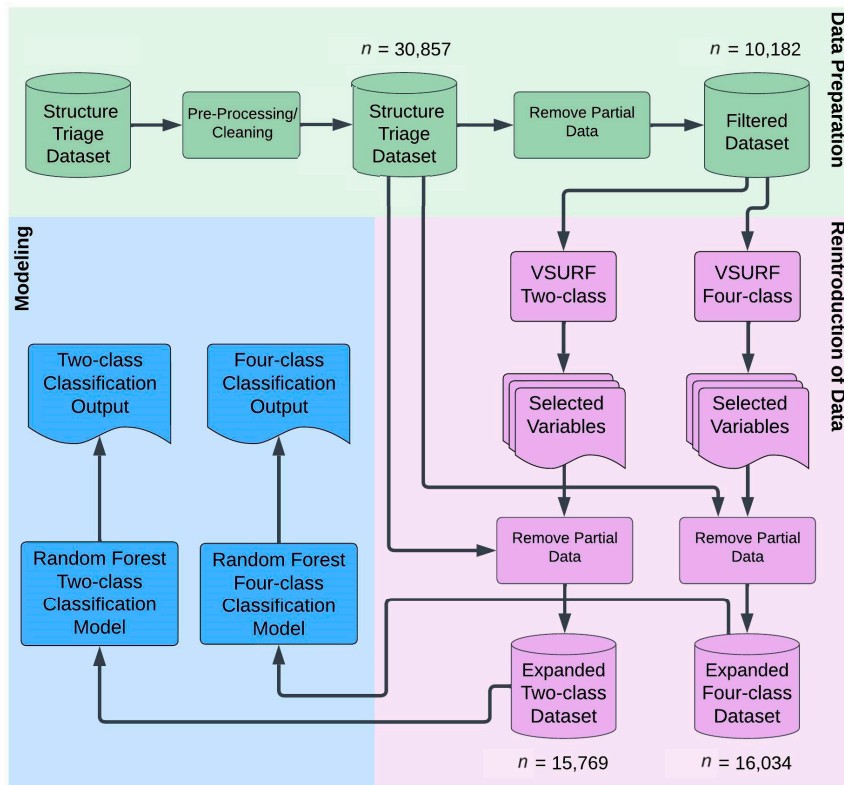

**Figure 5.** Modeling flowchart detailing the model creation process, data cleaning and preparation, reintroduction of data, and modeling.

The structure triage dataset contained 36,688 data points in the Western US with an assigned structure triage category. Preliminary data cleaning involved the correction or removal of misspelled or incorrectly labeled values, and removal of data points where the assigned structure triage category was "unknown". The resulting dataset contained 30,857 structure triage data points (Figure 5).

Since firefighters were not required to enter information into structure triage assessment fields, 20,675 out of 30,857 observations had values missing from one or more fields (Figure 6). For example, the lack of information about a structure could hinder the input of a value. If a firefighter were unaware of gas or how electricity was being supplied to a structure, they might skip that field. Time constraints might also have been a contributing factor; if a fire was moving quickly toward a community, it could be challenging to assess every structure fully.

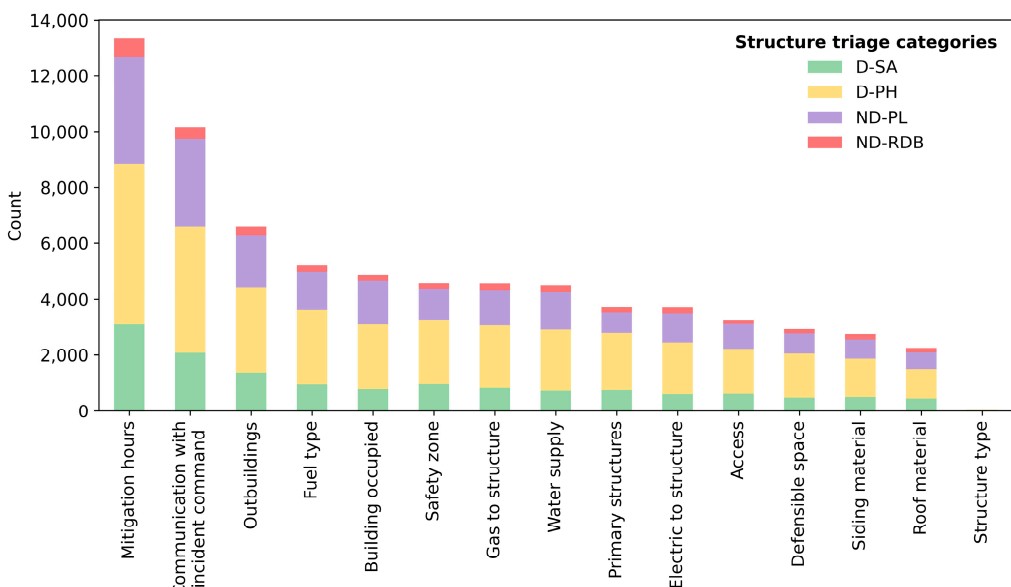

**Figure 6.** Bar chart illustrating the cumulative count of missing values in each variable by structure triage category.

Regardless of the reasons for leaving variables unassessed, missing data presented challenges as random forest models cannot be successfully constructed with data containing missing values. However, these data points may still have valuable information contained within their non-missing fields and excluding them may reduce the model's predictive capacity. In recognition of these issues, we developed a strategic approach to add partial data back into our modeling procedure.

Structure triage data points with any missing values were removed from the initial modeling process, resulting in a "filtered dataset" with a total of 10,182 data points (Figure 5). Following this, variable selection was performed on both the two-class and four-class datasets using the variable selection using random forests (VSURF) R package version 1.2.0 [48]. VSURF is a technique that uses random forests to assess the importance of each variable in the context of building a predictive model and selecting a subset of important variables from a dataset. The resulting subset of variables can then be used in subsequent modeling and analysis to potentially improve model performance or reduce dimensionality. Following variable selection, data points possessing values for all selected variables were added back into modeling to maximize the number of data points available for modeling (Figure 5). This added 5587 data points to the two-class classification model and 5852 data points to the four-class classification model.

The two-class and four-class models were evaluated using overall accuracy, precision, and recall. In addition to these metrics, ranger's permutation variable importance measure

was used to examine the variables with the greatest impact on predicting structure defensibility. Permutation variable importance measures the decrease in a model's performance when the values for a single predictor variable are randomly shuffled. When permuted, more important variables tend to negatively affect model performance, indicating that the model relies more heavily upon those variables [44]. Partial dependence plots were created using the pdp package version 0.8.1 [49] to gain a deeper understanding of the statistical relationships between the important variables and wildland firefighters' assessment of structure defensibility.

## 3. Results

### 3.1. Two-Class Classification Model

The variables selected for the two-class classification model included safety zone, mean percent cover of shrubs at a 6 km radius, road density at a 3.6 km radius, distance to major road type, slope at a 6 km radius, structure density at a 6 km radius, road density at a 6 km radius, access, mitigation hours, and defensible space. The model achieved an overall accuracy of 77.8% (Table 1). The defensible category exhibited a recall of 79.4%, accurately classifying 2000 observations while misclassifying only 520 as non-defensible. The precision for the defensible category was 83.4%, signifying its capability to accurately classify this category, with just 16.6% of predictions being incorrectly labeled as non-defensible. Conversely, the non-defensible category had a slightly lower recall of 75.4%. Out of the total observations, 1219 were correctly identified as non-defensible, with 398 being mistakenly classified as defensible. The non-defensible category demonstrated a precision of 70.1% and a relatively higher percentage of predictions being incorrectly labeled as defensible (30.0%). Overall, this model achieved high accuracy and demonstrated its ability to effectively classify structures as defensible, although its performance was slightly lower for non-defensible structures.

**Table 1.** Two-class classification model matrix with recall and precision accuracies. OA = overall accuracy.

| | | Actual Category | | Row Total | Precision |
|---|---|---|---|---|---|
| | | *Defensible* | *Non-Defensible* | | |
| **Predicted Category** | *Defensible* | 2000 | 398 | 2398 | 83.4% |
| | *Non-defensible* | 520 | 1219 | 1739 | 70.1% |
| | **Column Total** | 2520 | 1617 | 4137 | |
| | **Recall** | 79.4% | 75.4% | | **OA** = 77.8% |

Several variables emerged as important in predicting structure defensibility (Figure 7). Foremost, the presence of a safety zone was the most influential, with a permutation importance (indicating a change in model accuracy) of 16.1%. The mean percent cover of shrubs at a 6 km radius changed the model's performance by 7.5%. In terms of accessibility, variables such as the road density at a 3.6 km radius (7.2%), the road density at a 6 km radius (6.2%), and the distance to major roads (6.8%) were also found to be important. The slope at a 6 km radius and the structure density within a 3.6 km radius had similar permutation importance. Other contextual factors had lower permutation importance, with the access, mitigation hours, and defensible space variables each changing the accuracy by approximately 2–4%.

The partial dependence plots provide additional insights into the relationships between variables and defensibility (Figure 8). Structures with a potential safety zone were much more likely to be classified as defensible, underscoring its importance in determining structure defensibility (Figure 8a). Higher percentages of shrub cover at a 6 km radius were associated with an increased likelihood of structures being classified as non-defensible (Figure 8b). While the non-defensible category also had a higher probability at low shrub

cover, the rug plot shows few data points with low shrub cover, indicating that this specific relationship may not be reliable. The density of observations was higher between 25 and 50%, and Figure 8b reveals that the likelihood of non-defensible classification drastically increased at around 35% shrub cover.

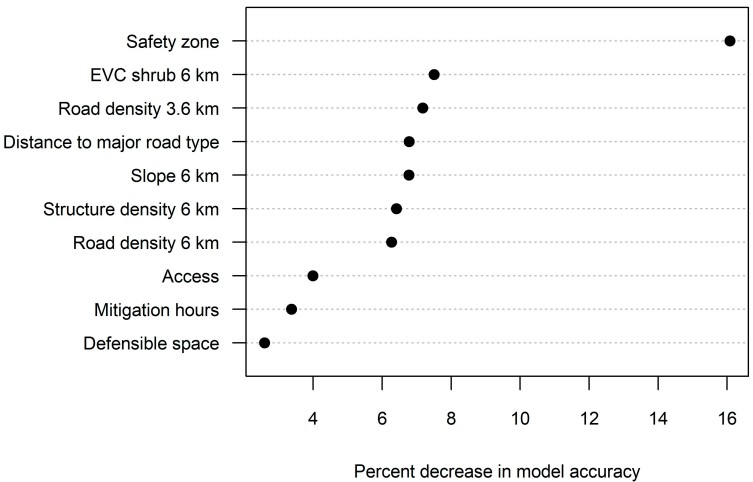

**Figure 7.** Variable importance using the permutation measure for the two-class classification model. The *x*-axis is the decrease in the model's accuracy when the values for the indicated single predictor variable were randomly shuffled in the random forest modeling process.

As the distance of a structure from a major road increased, the chance of a structure being classified as defensible decreased (Figure 8d). Regarding the slope at 6 km, structures were more likely to be deemed defensible at lower slope degrees (Figure 8e). The plot for access indicates that structures capable of accommodating larger wildland firefighting equipment with significant water-holding capacities were more likely to be classified as defensible (Figure 8h). This can be attributed to larger equipment and water storage enhancing firefighting effectiveness, thereby increasing the likelihood of structures being deemed defensible.

There were noteworthy associations between mitigation hours, the presence of defensible space, and the likelihood that structures would be classified as defensible or non-defensible (Figure 8i,j). As the number of mitigation hours increased, structures were more likely to be classified as non-defensible. Structures labeled as having "No defensible space" were considerably more likely to be classified as non-defensible. In contrast, the road density at a 3.6 km and 6 km radius (Figure 8c,g), and the structure density at a 6 km radius (Figure 8f) had relatively small impacts on the likelihood of structures being categorized as non-defensible or defensible, making it challenging to draw definitive conclusions for how these factors may impact the assessment of structure defensibility.

### 3.2. Four-Class Classification Model

The selected variables for the four-class model included safety zone, mitigation hours, longitude, latitude, mean percent cover of trees at a 6 km radius, structure density at a 3.6 km radius, mean percent cover of shrubs at a 6 km radius, and defensible space. The model's accuracy decreased relative to the two-class model from 77.8% to 62.4% (Table 2). Some notable patterns emerged when evaluating the model's performance in individual structure triage categories. D-PH achieved a recall rate of 74.3% along with a precision rate of 60.4%. This implies that the model often overpredicted D-PH, which is unsurprising given its high prevalence in the dataset. This frequency of overpredicting indicates a propensity for commission errors, where the model tends to overclassify instances of D-PH, leading to a decrease in the precision.

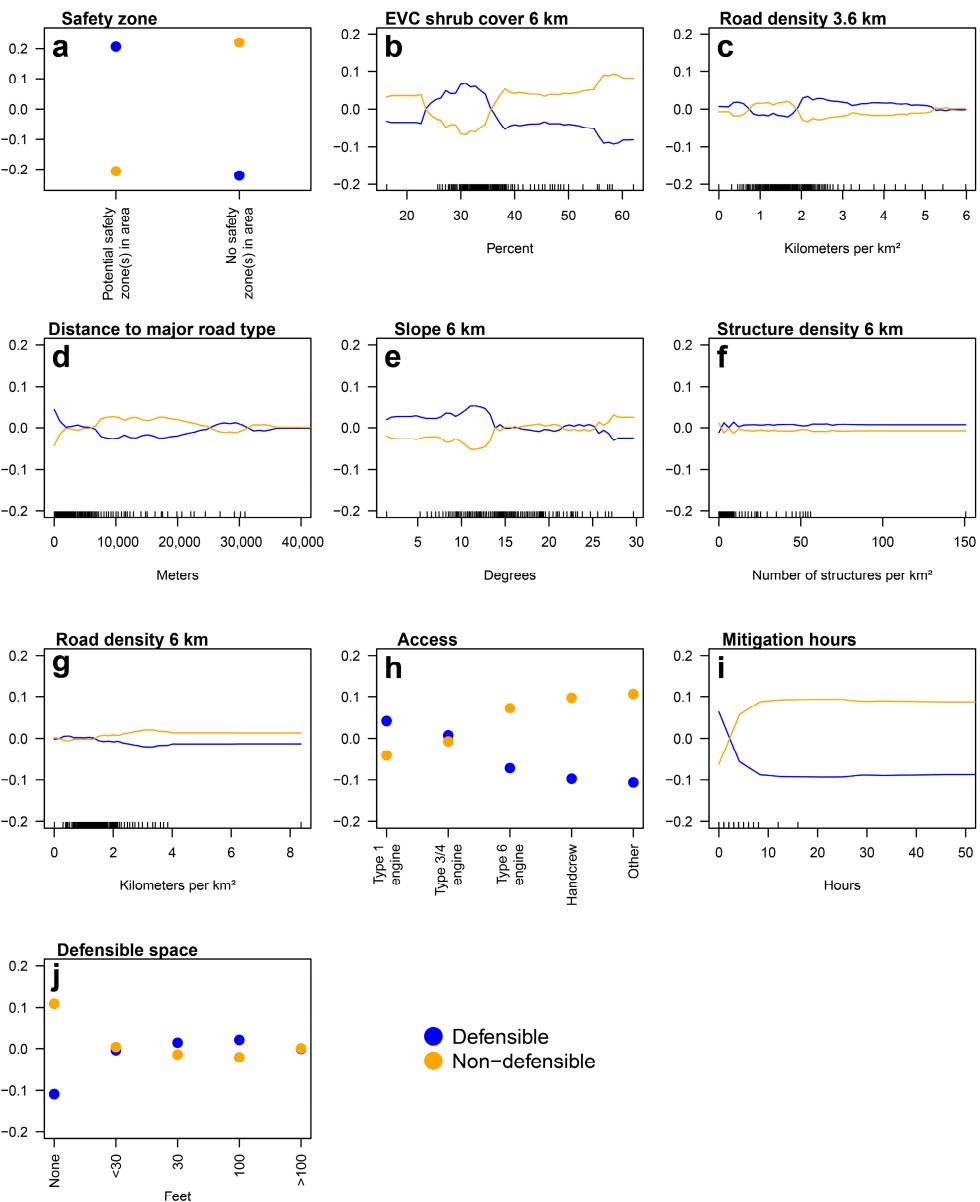

**Figure 8.** Partial dependence plots showing the normalized average predicted probability of each variable in the two-class classification, calculated by subtracting the proportion of each class in the training dataset from the corresponding PDP values, effectively centering the PDP results relative to the class distribution. For continuous variables, the density of data points is represented along the *x*-axis with a "rug plot". Plots (**a**–**j**) are arranged in descending order based on their relative variable importance.

In contrast, ND-PL exhibited a reasonably balanced accuracy, with a recall accuracy of 65.1% and a precision rate of 65.5%, indicating similar occurrences of omission and commission errors. The remaining 35% of the classification challenge mainly arose from the model's tendency to overpredict D-PH, leading to an increased number of false positives for D-PH, resulting in confusion between these two categories.

For the D-SA category, the situation was quite the opposite: D-SA is a rarity in the dataset, and it was notably underpredicted, as reflected by the recall accuracy of 34% and the precision rate of 61.3%. This phenomenon may be attributed to the behavior of random forests, where there is a tendency to overemphasize abundant classes while downplaying rare ones [50,51]. This phenomenon is closely linked to the bagging process, which involves randomly selecting subsets of the training data to build individual decision trees. Given

that only a fraction of the data is used for each tree, rare classes may have a limited presence in these subsets. Consequently, this can result in a decreased influence of rare classes on the individual decision trees, causing the overall model to struggle when faced with such infrequent instances [50,51].

**Table 2.** Four-class classification model matrix with recall and precision accuracies. OA = overall accuracy.

|  |  | Actual Category | | | | | |
|---|---|---|---|---|---|---|---|
|  |  | *D-SA* | *D-PH* | *ND-PL* | *ND-RDB* | **Row Total** | **Precision** |
| **Predicted Category** | *D-SA* | 192 | 105 | 8 | 8 | 313 | 61.3% |
|  | *D-PH* | 295 | 1266 | 481 | 54 | 2096 | 60.4% |
|  | *ND-PL* | 80 | 332 | 911 | 67 | 1390 | 65.5% |
|  | *ND-RDB* | 1 | 0 | 0 | 1 | 2 | 50.0% |
|  | **Column Total** | 568 | 1703 | 1400 | 130 | 3801 |  |
|  | **Recall** | 33.8% | 74.3% | 65.1% | 0.8% |  | **OA** = 62.4% |

In the case of the ND-RDB category, the model displayed a dismal recall accuracy of 0.8% and a low precision rate of 50%. It is crucial to recognize that the 50% precision should be approached with caution. This is because the model made merely two predictions pertaining to ND-RDB instances. When working with such a limited number of predictions, the presence of one correct prediction does not offer a reliable performance measure.

Several variables appeared to be important when predicting structure defensibility using all four structure triage categories (Figure 9). Notably, the presence of a safety zone emerged as the most influential, leading to a 16.2% change in the model accuracy. Longitude and latitude also caused substantial changes to the model's performance by 14.1% and 11.7%, respectively, indicating potential geographic differences in the assessment of structure triage classes. The vegetation variables, specifically the mean percent tree and shrub cover within a 6 km radius, influenced the model's performance by 10.7% and 9.3%, respectively. Structure density within a 3.6 km radius had a similar permutation importance. Lastly, the number of mitigation hours had a 7.9% impact on model accuracy, while defensible space brought about a 4.3% change in the model's performance. While the importance was higher for many variables selected for the four-class model, we note that the model itself was less capable of distinguishing between classes in comparison to the two-class model. The partial dependence plots for the four-class classification are shown in Appendix A, Figure A1, but should be interpreted with caution given the relatively poor model performance and the less prevalent classes (D-SA and ND-RDB) being poorly accounted for by the model.

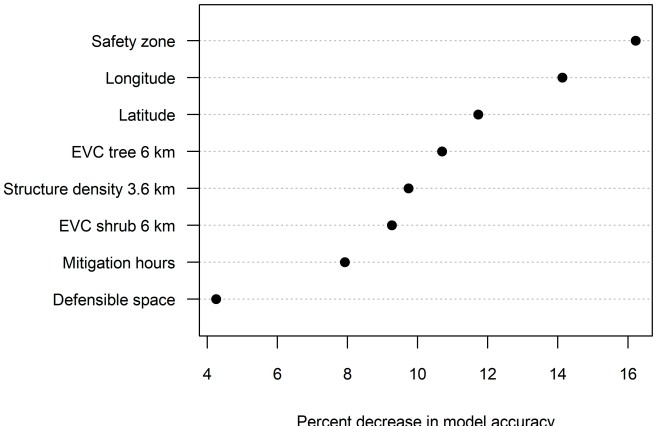

**Figure 9.** Variable importance using the permutation measure for the four-class classification model. The *x*-axis is the decrease in the model's accuracy when the values for the indicated single predictor variable were randomly shuffled in the random forest modeling process.

## 4. Discussion

In our two-class classification model, we achieved a high level of accuracy, coupled with high recall and precision accuracies. However, the four-class model exhibited lower prediction accuracy. This was primarily due to an imbalance in the dataset, where the D-PH and ND-PL categories accounted for 81% of the training data. As a result, the model struggled to classify the rarer D-SA and ND-RDB categories accurately. As the precision of the classification scheme increased, transitioning from a two-class to a four-class model, it is conceivable that individual-level assessments may exhibit a heightened degree of ambiguity and subjectivity among different individual cases. In other words, as the granularity of the classification system intensifies, the distinctions between categories become more subtle, potentially resulting in varying and less clear-cut evaluations made by different individuals. Focusing solely on distinguishing between two outcomes simplifies and streamlines the decision-making process, eliminating some nuanced differences when assessing a structure's defensibility.

In addition to creating predictive models, this study also provided insights into how firefighters may weigh different variables during their evaluation of structure defensibility. The importance of proximal safety zone presence in both models emphasizes the high importance firefighters likely place on maintaining their own safety and how this consideration affects their determination of structure defensibility. It is worth noting that the presence of a safety zone is dependent on many factors, including environmental conditions, wildfire behavior, and individual judgment. Recent efforts to employ geospatial techniques for mapping and providing quantitative descriptions of safety zones offer one avenue for making safety zone assessments less subjective [15,52,53].

The presence of nearby major roads also appears to be an important determinant of structure defensibility. Roads provide convenient, rapid access for firefighters and emergency response teams. This accessibility can enable faster response times and facilitate the efficient transportation of firefighting equipment. Recognizing the importance of these road networks in enhancing defensibility can inform better urban planning and emergency preparedness strategies, contributing to safer and more resilient communities.

Variables related to vegetation, such as the average percentage of shrub cover, were also found to be important in the two-class and four-class classification models. Vegetation is the primary means by which wildfire propagates. High fuel loads can significantly increase the danger posed to structures by increasing the intensity and spread of wildland fire [3]. Both the two- and four-class classification models included shrub cover at a 6 km radius as a variable, while the four-class model also included tree cover at a 6 km radius. The two-class classification model indicated that elevated shrub cover at a landscape scale heightens the likelihood of structures being classified as non-defensible. Interestingly, a notable sharp increase in non-defensibility was observed at around 35% cover. Our findings emphasize the importance of landscape scale vegetation, which aligns with previous research conducted by Syphard et al. [26] and Braziunas et al. [25], supporting the notion that landscape scale vegetation may serve as a better predictor of structure loss compared to local scale vegetation.

The mean slope within a 6 km radius was selected as a variable in the two-class model. The partial dependence plot in Figure 8e shows that structures in areas with shallow slopes were more likely to be categorized as defensible. This finding is consistent with prior research indicating that structures on steeper slopes are more susceptible to loss or destruction during wildfires [16,17,30]. However, variables capturing slope at finer scales were not selected, indicating that landscape scale slope may have similar value as landscape scale vegetation cover for explaining structure defensibility.

Although this study provides valuable insights into the complex relationships between variables and firefighter assessments of structure defensibility, it is important to acknowledge certain limitations in the research methodology. One notable limitation is the reliance on subjective assessments by individual firefighters during the data collection

process. Determining structure defensibility is inherently subjective and can vary based on each firefighter's experience, training, and personal judgment.

Examining Figure 6 reveals a consistent pattern in the completion of variables, with structure type, roof material, siding material, defensible space, and access being the most reliably filled out. These variables, arguably less subjective and easier to observe and document, stand out in their consistency. In contrast, variables such as electricity to structure, primary structure, water supply, gas to structure, safety zone, building occupancy, and mitigation hours are more difficult to ascertain. These variables tend to be more subjective or are not as easily observable, which could contribute to the more frequent omission of these variables in the dataset, reflecting the inherent complexity involved in their assessment. In the modeling process, a notable distinction arose between the structure triage variables and the GIS-derived variables. With the exception of the very important safety zone variable, the structure triage variables were generally found to be less important than the GIS-derived variables. Many of the structure triage variables required subjective judgment, and external factors or inconsistencies in the assessment process might have influenced their values.

Firefighters' decision-making in structure triage may show confirmation bias, where they unconsciously prioritize information aligning with their beliefs, downplaying contradictory data. This cognitive process can lead to firefighters giving greater importance to specific variables, potentially at the expense of others, to validate their initial assessments. The adoption of a balanced evaluation approach in structure triage could prove beneficial, and further research in this field may provide valuable practical insights.

Other limitations include the lack of specific information about fire behavior, weather conditions, time constraints, and other factors associated with each fire incident. This information could improve the understanding of the context in which the assessments of defensibility were made. For instance, a structure may have been considered defensible during a period with lower wind speeds, but non-defensible during a period with anticipated higher wind speeds. Despite these limitations, our predictive capabilities maintained a high level of accuracy. However, considering these factors could provide additional insights into the effectiveness of a variety of strategies and tactics used to protect structures and further enhance predictive accuracy.

Our findings have practical implications for fire management practices. One potential application is enhancing structure triage training for fire personnel. Incorporating the study's findings on which variables are most closely associated with structure defensibility can provide valuable guidance for assessing risks to structures during wildfires, and may be useful for creating heuristics that increase the efficiency in assessing defensibility. Our findings also have implications for communities. By collaborating with fire management professionals, communities can develop comprehensive wildfire response plans and implement measures to mitigate fire risks to improve the likelihood of structures in that community being categorized as defensible. For example, ensuring that safety zones are available close to structures in a community would seem to be critical for increasing the likelihood of structures being assessed as defensible. Additionally, maintaining and clearing access roads leading to properties within a community may improve their perceived defensibility. These actions can be combined with commonly advocated maintenance of defensible space around structures [9,22,25]. By leveraging these findings, communities can develop informed strategies and implement measures that enhance the likelihood of structures being categorized as defensible, thereby optimizing risk mitigation and contributing to overall wildfire resilience within the WUI.

Future research holds considerable potential to advance structure triage practices by applying spatial analysis and modeling techniques. By incorporating GIS, remote sensing, and machine learning, researchers could develop models that enhance the accuracy and effectiveness of identifying structures' defensibility. For example, by harnessing these advanced technologies, researchers could create spatial models that incorporate additional variables and factors, such as structure location, vegetation information, long-term simu-

lations of wildfire behavior, fuel models, and fire weather. By integrating these data and modeling techniques, researchers have the potential to develop a comprehensive national scale map that offers more accurate and detailed assessments of structures' predicted defensibility levels. This type of map could provide a valuable tool to enable decision-makers, emergency responders, and land managers to visualize and assess the vulnerability of structures in communities.

Similarly, examining the correlation between structure triage ratings obtained during an incident and the subsequent structure loss in wildfires offers a promising direction for future research. The empirical validation process could compare structure triage ratings, on-the-ground protection measures implemented by firefighters, and the survival or loss of structures. This type of analysis could serve to provide an improved understanding of the accuracy and value of structure defensibility assessments in the dynamic context of actual wildfire events.

## 5. Conclusions

The main focus of this research was to advance our understanding of structure triage and defensibility by developing robust predictive models that provide insight into the factors that may be considered by firefighters during structure triage assessment. This study was the first to utilize a structure triage dataset maintained by NIFC in a predictive modeling context, and demonstrates the effectiveness of classifying structures as either defensible or non-defensible based on the considered variables and criteria. This study also highlights the challenges of making more precise predictions that further distinguish between defensible and non-defensible classes in a four-class classification model. Our models highlighted the extreme importance of safety zones for the assessment of structure defensibility, where the absence thereof may play a strong role in a firefighter deeming a structure non-defensible. GIS-derived landscape scale vegetation and road variables, which were not included in the structure triage dataset, but are believed to be important predictors of defensibility, were also consistently found to be important across both the two and four-class models. These findings will contribute to improving the consistency and reliability of the structure triage process. They also provide unprecedented insights into the importance of firefighter safety in making fire management decisions. Lastly, the findings are useful more broadly to anyone who may live in fire-prone WUI settings, as they provide direct evidence towards the structural and landscape conditions that may influence firefighters' determinations of whether or not a structure can be safely and effectively defended.

**Author Contributions:** Conceptualization, A.J.H., P.E.D. and M.J.C.; methodology, A.J.H., P.E.D. and M.J.C.; software, A.J.H.; formal analysis, A.J.H.; resources, P.E.D., M.J.C. and M.P.T.; data curation, A.J.H.; writing—original draft preparation, A.J.H.; writing—review and editing, A.J.H., P.E.D., M.J.C. and M.P.T.; visualization, A.J.H.; supervision, P.E.D. All authors have read and agreed to the published version of the manuscript.

**Funding:** This research was funded by the USDA Forest Service (agreement number 21-CS-11221636-120).

**Institutional Review Board Statement:** Not applicable.

**Informed Consent Statement:** Not applicable.

**Data Availability Statement:** Data were obtained from the National Interagency Fire Center (NIFC). Restrictions apply to the availability of these data.

**Conflicts of Interest:** The authors declare no conflict of interest.

## Appendix A

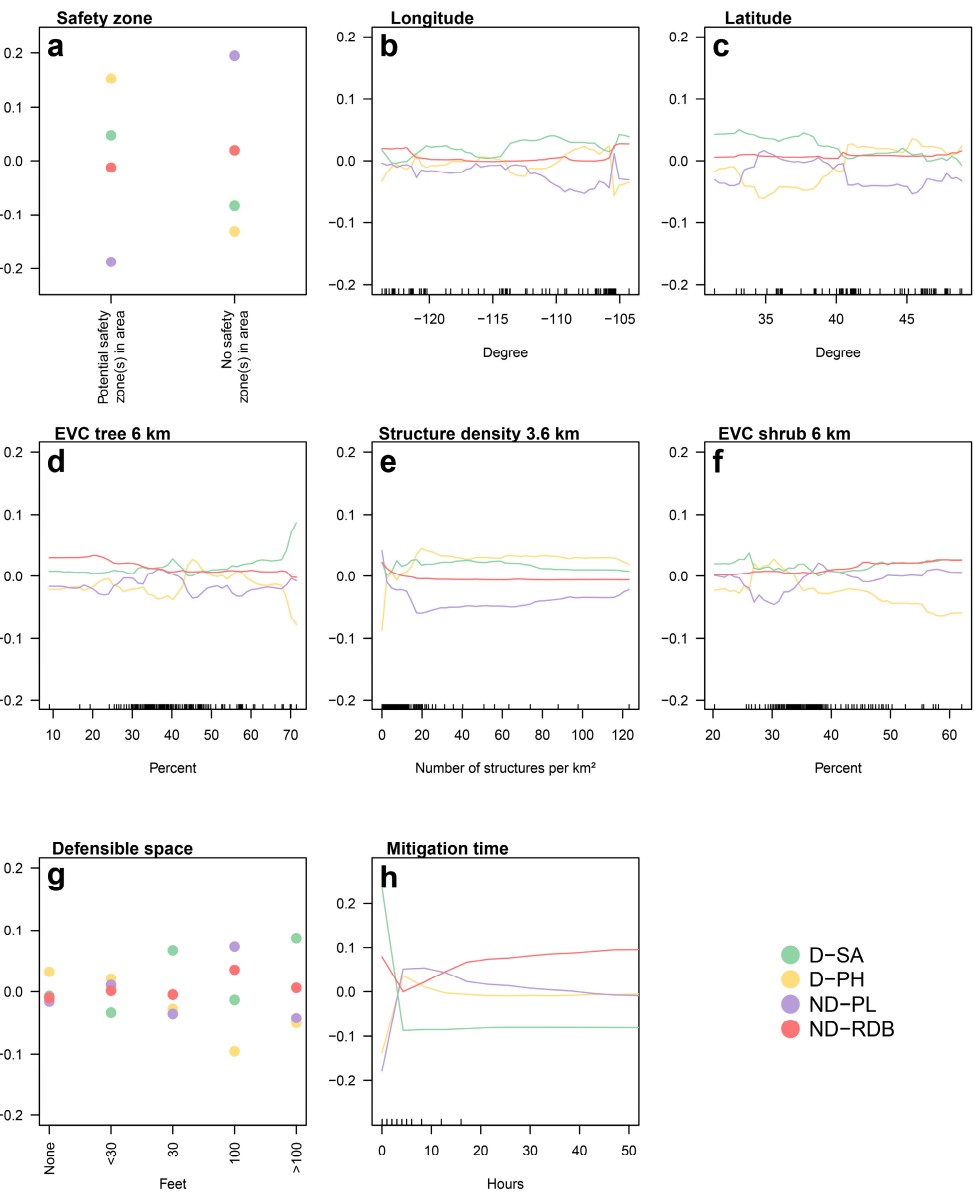

**Figure A1.** Partial dependence plots showing the normalized average predicted probability of each variable in the four-class classification, by subtracting the proportion of each class in the training dataset from the corresponding PDP values, effectively centering the PDP results relative to the class distribution. For continuous variables, the density of data points is represented along the *x*-axis with a "rug plot". Plots (**a**–**h**) are arranged in descending order based on their relative variable importance.

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
