# Peer review of "Modeling Wildland Firefighters’ Assessments of Structure Defensibility"

_fire, doi:10.3390/fire6120474_

Round 1

Reviewer 1 Report

Comments and Suggestions for Authors

The presented article deals with the issue of “modeling wildland firefighters’ assessments of structure defensibility”. The Abstract briefly and concisely summarizes content of the article and provides basic information. The chapter "Introduction“ informs the reader about the given research topic in sufficient detail. The applied methodology of the implemented scientific-research procedures is described in detail within the second chapter "Materials and Methods", in the framework of its four subsections. The utilised procedures are chosen correctly and suitably in order to achieve the defined goals of the given research activities. The achieved results are interesting. They are presented in the broad area of the third chapter, which is followed by a comprehensive discussion in the fourth chapter. The illustrative figures and graphs complement the text appropriately and comprehensibly. The list of references is also extensive, with the amount of items 53. I emphasize high-quality processing of the article from both a professional and a formal point of view. The authors are evidently experts in the given professional area. The article meets the criteria required for publication of a scientific research work in the given professional journal. I recommend publishing of it in the presented form, without notes or recommendations.

Author Response

We thank Reviewer 1 for their comments and their review of our paper.

Reviewer 2 Report

Comments and Suggestions for Authors

Dear Authors,

The paper created a machine learning model that can enhance comprehension of elements contributing to evaluated structural defensibility using a dataset containing triage assessments of tens of thousands of structures in the Western US. In addition to the variables gathered by wildland firefighters—such as the surrounding ignition zone and structural characteristics—the random forest models also made use of landscape variables that captured crucial details about topography, vegetation, accessibility, and structure density that were not included in the triage dataset. Structures were classified as defensible or non-defensible in binary classification with a high overall accuracy of 77.8%, and it is discovered that the presence of a safety zone was the most crucial criterion in evaluating a structure's defensibility. It was also shown that terrain, vegetation type, and road proximity were highly significant. The manuscript managed to present novel matter in the field, however the authors should consider some minor revisions before publication.

- For better reading, think about segmenting the sentences in the abstract and introduction. Break the long sentences to some short ones. Abstract should be a general view of the manuscript briefly. So, edit the abstract.

-There are some English language errors that should be addressed.

-Explain about the originality and accuracy of the dataset.

-In what base do you choose the variables? It is a little vague. Clarify it.

-Why do you plummet the dataset using pre-processing? Many of the datasets missed.

-Conclusion should be changed and shortened. Just express the main achievements there.

-There are some self-cited references. Do your best to change the references.

Regard

Comments on the Quality of English Language

There are some English language errors that should be addressed.

Author Response

We thank Reviewer 2 for their helpful comments. Our responses to the reviewer’s concerns are listed in blue below. Line numbers refer to the document with tracked changes shown. 

For better reading, think about segmenting the sentences in the abstract and introduction. Break the long sentences to some short ones. Abstract should be a general view of the manuscript briefly. So, edit the abstract.

We edited the abstract, and broke two long sentences into four shorter sentences (lines 14-20):

Our random forest models utilized variables collected by wildland firefighters, including structural characteristics and the surrounding ignition zone. The models also used landscape variables not contained within the triage dataset that captured important information about accessibility, vegetation, topography, and structure density. We achieved high overall accuracy (77.8%) in classifying structures as defensible or non-defensible. Presence of a safety zone was the most important factor in determining structure defensibility.

There are some English language errors that should be addressed.

We carefully read through the manuscript and made language edits.  

Explain about the originality and accuracy of the dataset.

We added a statement on originality to Section 2.1 lines 97-98:

To our knowledge, this is the first published analysis of the structure triage dataset.

We also added a statement on originality to the Conclusions (lines 566-567):

This study was the first to utilize a structure triage dataset maintained by NIFC

The accuracy of the dataset is addressed on lines 492-497 and 512-517:

Although this study provides valuable insights into the complex relationships between variables and firefighter assessments of structure defensibility, it is important to acknowledge certain limitations in the research methodology. One notable limitation is the reliance on subjective assessments by individual firefighters during the data collection process. Determining structure defensibility is inherently subjective and can vary based on each firefighter’s experience, training, and personal judgment.

Firefighters’ decision-making in structure triage may show confirmation bias, where they unconsciously prioritize information aligning with their beliefs, downplaying contradictory data. This cognitive process can lead to firefighters giving greater importance to specific variables, potentially at the expense of others, to validate their initial assessments. The adoption of a balanced evaluation approach in structure triage could prove beneficial, and further research in this field may provide valuable practical insights.

In what base do you choose the variables? It is a little vague. Clarify it.

We have added the following sentences to Section 2.2 (lines 136-139):

Analysis variables were selected based on variables included in the structure triage dataset, plus additional GIS-derived variables describing vegetation, topography, structure density, and road characteristics. Previous work has demonstrated relationships between similar GIS-derived variables and structure loss [16,17,25,26,30].

Why do you plummet the dataset using pre-processing? Many of the datasets missed.

We apologize that our previous description of preprocessing was confusing. Many of the data points in the structure triage dataset only contained structure locations, and did not have a structure triage category (and thus could not be modeled). We have simplified our description of the pre-processing steps, and hope this explanation has improved clarity (lines 281-385):

The structure triage dataset contained 36,688 data points in the Western US with an assigned structure triage category. Preliminary data cleaning involved the correction or removal of misspelled or incorrectly labeled values, and removal of data points where the assigned structure triage category was “unknown”. The resulting dataset contained 30,857 structure triage data points (Figure 5).

Conclusion should be changed and shortened. Just express the main achievements there.

We have moved the future research paragraphs previously in the Conclusions to the Discussion section, and the Conclusion (lines 564-5582) now just briefly summarizes the main achievements of this research.

There are some self-cited references. Do your best to change the references.

We have reviewed self-citations, and found that they are relevant to key points made in the manuscript. 

Reviewer 3 Report

Comments and Suggestions for Authors

Advantages.

Good practical research in the field of wildland fires analysis using random forest learning model.  The experimental part is impressive. The topic of this article is interesting and meaningful for precise prediction and control of firefighter decision-making. 

The design of the manuscript is well structured:

-          Introduction part is given (with literature analysis).

-          The methodology part with data description is given (Materials and Methods).

-          Experimental results and analysis part is given.

-          Discussion and conclusion parts are given.

-          References to literature, figures, and tables are correct.

There are no significant criticisms about the research methodology – random forest method is optimal for this research.

Disadvantages:

-          Is not justified the random forest method choosing.

-          You use a classification model, the output of the random forest is the class selected by most trees.

-          It is acceptable that the “reduced dataset” contains 30857 of 71925 data points? (Line 227)

-          Figure A1 is very similar to Figure 8. Why is it not in the main text?

Author Response

We thank Reviewer 3 for their helpful comments. Our responses to the reviewer’s concerns are listed in blue below. Line numbers refer to the document with tracked changes shown. 

Is not justified the random forest method choosing.

We have added a short section of text at the beginning of Section 2.4 justifying our use of  random forests (lines 263- 269):

Random forests is a machine learning method that constructs a collection of decision trees by building each tree with a random subset of the training data, and the predictions from these trees are aggregated to produce a final prediction [44]. This method is ideal for this analysis for several reasons, including supporting both binary and multi-class classification tasks, the capability to handle both categorical and continuous predictors, being able to characterize non-linear and interactive relationships between predictor and response variables, and providing robust measures of variable importance.

You use a classification model, the output of the random forest is the class selected by most trees.

We added a clarifying sentence describing how random forests work on lines 263-265:

Random forests is a machine learning method that constructs a collection of decision trees by building each tree with a random subset of the training data, and the predictions from these trees are aggregated to produce a final prediction [44].

It is acceptable that the “reduced dataset” contains 30857 of 71925 data points? (Line 227)

We apologize that our previous description of preprocessing was confusing. Many of the data points in the structure triage dataset only contained structure locations, and did not have a structure triage category (and thus could not be modeled). We have simplified our description of the pre-processing steps, and hope this explanation has improved clarity (lines 281-285):

The structure triage dataset contained 36,688 data points in the Western US with an assigned structure triage category. Preliminary data cleaning involved the correction or removal of misspelled or incorrectly labeled values, and removal of data points where the assigned structure triage category was “unknown”. The resulting dataset contained 30,857 structure triage data points (Figure 5).

Figure A1 is very similar to Figure 8. Why is it not in the main text?

The four-class classification had lower accuracy, and largely failed to predict the ND-RDB class. For these reasons, we think the partial dependence plots should be interpreted with caution. We have indicated this in the manuscript on lines 435-438:

Partial dependence plots for the four-class classification are shown in Figure A1, but should be interpreted with caution given relatively poor model performance and the less prevalent classes (D-SA and ND-RDB) being poorly accounted for by the model.